# Hyperbranched Polyglycerols as Robust Up-Conversion Nanoparticle Coating Layer for Feasible Cell Imaging

**DOI:** 10.3390/polym12112592

**Published:** 2020-11-04

**Authors:** Mingcong Hao, Tongtong Wu, Qunzhi Chen, Xueyan Lian, Haigang Wu, Bingyang Shi

**Affiliations:** School of Life Sciences, Henan University, Kaifeng 475003, China; haomc@vip.henu.edu.cn (M.H.); 1014753190785@vip.henu.edu.cn (T.W.); cqz723902674@163.com (Q.C.); l17839193922@163.com (X.L.)

**Keywords:** hyperbranched polyglycerols, up-conversion nanoparticle, bioimaging

## Abstract

Owing to the wide spectrum of excitation wavelengths of up-conversion nanoparticles (UCNPs) by precisely regulating the percentage of doping elements, UCNPs have been emerging as bioimaging agents. The key drawback of UCNPs is their poor dispersibility in aqueous solution and it is hard to introduce the chemical versatility of function groups. In our study, we present a robust and feasible UCNP modification approach by introducing hyperbranched polyglycerols (hbPGs) as a coating layer. When grafted by hbPGs, the solubility and biocompatibility of UCNPs are significantly improved. Moreover, we also systematically investigated and optimized the chemical modification approach of amino acids or green fluorescence protein (GFP), respectively, grafting onto hbPGs and hbPGs-*g*-UCNP by oxidizing the vicinal diol to be an aldehyde group, which reacts more feasibly with amino-containing functional molecules. Then, we investigated the drug-encapsulating properties of hbPGs-Arg with DOX and cell imaging of GFP-grafted hbPGs-*g*-UCNP, respectively. The excellent cell imaging in tumor cells indicated that hbPG-modification of UCNPs displayed potential for applications in drug delivery and disease diagnosis.

## 1. Introduction

The application of nanomaterials in disease diagnostics and therapeutics has been a mostly attractive frontier in biomaterials—for example, micelles as drug delivery carriers [1], and nanoparticles as tumor issue imaging reagents [2,3]. Surface engineering modifications on bio-nano interfaces will significantly improve the biocompatibility and targetability of nanomaterials [4,5]. However, how to quickly and feasibly fabricate surface-functionalized nanocarriers has been the most essential work for decorating nanocarriers, especially for improving the biocompatibility and easily grafting the functional groups, such as peptides [6,7] and proteins [8,9]. To develop such a technique will be helpful to further extend the application in related fields of nanomaterials.

More recently, hyperbranched polyglycerols (hbPGs) have been seen as one of the most attractive hyperbranched polymers [10], which is owed to their excellent biocompatibility [11,12] and feasibly surface-initiating grafting method [13,14,15]. Through the hbPG-decorating process, the water solubility and dispersity of nanoparticles can be significantly improved, which is more feasible than other processes—e.g., diphosphate-PEG-grafted up-conversion nanoparticles (UCNPs) [8]. UCNPs, especially, have been widely utilized in cell imaging [16], photodynamic therapy [17] and drug delivery [18]. However, synthesis of functionalized hbPGs requires a rigorous reacting condition—for example, carboxyl groups under high temperature [19] and amino with multiple reaction steps [20]—which is not a moderate biological condition. Developing a feasible and moderate biological method can extend hbPG applications in the concerned field dramatically. As several previous works in the literature have reported [21,22,23,24], the glycose moiety or vicinal diol can be easily and moderately oxidized to form an aldehyde group, which is useful for modifying amino-terminated biomolecules. Moreover, the aldehyde-containing hbPGs will be cross-linked owing to the hemiacetal formation [20]. As a result, the development of an amino-functionalized protocol in aqueous solution will be more attractive for preparing biomolecule-labeled nanocarriers.

In this report, we investigated one aqueous reductive amination method based on aldehyde-terminated hbPGs, which was obtained by using NaIO_4_ as an oxidation reagent. Firstly, we used an NMR technique to characterize the oxidation product in aqueous solution, which indicated that the aldehyde-terminated hbPGs did not take part in the cross-linking process to form a hemiacetal. Then, we utilized these aldehyde-terminated hbPGs to form arginine-terminated hbPGs and green fluorescence protein (GFP)-grafted hbPGs-*g*-UCNP. The corresponding biological properties of these products, e.g., zeta potential and cytotoxicity, will test whether an aldehyde-mediated functionalization process is suitable for surface engineering of hbPGs shells.

## 2. Experimental Part

### 2.1. Chemicals and Materials

Glycidol (96%), NaIO_4_ (99%), D_2_O (99.8%) were purchased from J&K company (Shanghai, China); 4 Å sieve, NaH in mineral oil (40% w/w), NaBH_4_, arginine (95%), Doxorubicin hydrochloride (98%) were purchased from Aladdin company (Shanghai, China). 1,4-Dioxane (AR) was bought from Lingfeng company (Shanghai, China). All the chemical reagents were not purified unless specified. A549 and U87MG cell lines were obtained from the Chinese Academy of Science (Shanghai, China).

### 2.2. Synthesis of hbPGs

Before the synthesis experiment, all the solvent was dried by using a 4 Å sieve. Hydroquinone (1 mmol) was dissolved in 25 mL of anhydrous 1,4-dioxane, and then this mixture was bubbled by N_2_ gas for 30 min to remove dissolved oxygen gas. Therefore, NaH in mineral oil (0.1 eq.) was added to the mixture under N_2_ ambient conditions and the mixture was heated in an oil bath at 95 °C for 30 min. After, glycidol (5 mL) in 20 mL of anhydrous 1,4-dioxane was slowly added to the reaction solution and further stirred for another 12 h after the addition process. When the reaction finished, the reaction mixture was cooled to room temperature and the top organic layer was poured out. The residual gel-like products were dissolved in 10 mL of de-ionized water and purified by dialysis bag (molecular weight cutoff > 3.5 kDa). The product solution was freeze-dried to obtain a brown-yellow gel product and stored at 4 °C for the next step.

### 2.3. Reductive Amination

Reduction amination process in methanol was performed as follows: hbPGs (300 mg) was dissolved in methanol and NaIO_4_ (150 mg) was added to the methanol solution. This reaction mixture was vigorously stirred until that NaIO_4_ crystal became powder condition. After the reaction finished, the crude production solution was centrifugated to remove inorganic salt at 4 °C (5000 rpm for 5 min). Thereafter, arginine (125 mg, 1 eq.) was added to the mixture and stirred for another 24 h. Therefore, NaBH_4_ (106 mg, 4 eq.) was slowly added to the mixture and stirred for another 4 h. After the reaction finished, the methanol was removed by using an evaporator and redissolved in water (10 mL). The crude products were purified by dialysis bag, the same as the aforementioned process. The final product in aqueous solution was freeze-dried to obtain the Arg-terminated hbPGs.

### 2.4. Preparation of hbPGs Grafting UCNP and Further Modification Process

The UCNP hexane solution was added to the three neck bottles and it was blow-dried to obtain the dried UCNP powder. Then, 10 mL of anhydrous 1,4-dioxane was added to suspend the nanoparticles. In total, 10 mg of NaH in oil mineral was added to the dioxane suspended solution and the mixture was heated at 95 °C for 30 min. Glycidol (5 mL) in 20 mL of anhydrous 1,4-dioxane was slowly added to the suspended solution and vigorously stirred, which was same as the hbPG synthesis process. After the reaction finished, the reaction mixture was cooled down to room temperature and the organic layer was poured out. The gel-like product was dissolved in 20 mL of de-ionized water, and this suspended solution was centrifugated to obtain the hbPGs-*g*-UCNP products (15000 rpm for 5 min). These crude products were purified by using a dissolving and centrifugation process three times. Then, the final product was freeze-dried to obtain anhydrous product powder and stored at 4 °C for further modification process.

HbPGs-*g*-UCNP (5.0 mg) was re-suspended in deionized water (500 μL) and 100 μL of NaIO_4_ (10 mg/mL) was added to the mixture, which was the essential step to get the aldehyde-terminated UCNP nanoparticles. This suspended solution was kept at 37 °C for 12 h, and then this mixture was centrifugated to separate the hbPGs-g-UCNP from the reacting solution. This crude product was purified by using a suspending and centrifugation process three times. At last, the moisture UCNP was freeze-dried and stored at 4 °C for the next step.

HbPGs-*g*-UCNP-CHO (5.0 mg) was re-suspended in deionized (DI) water, and then 100 μL of GFP protein solution was added into this mixture to be further stirred for 12 h. After the reaction finished, NaBH_4_ (10 mg) was added to the mixture and stirred for another 4 h. At last, the hbPGs-*g*-UCNP-GFP was centrifugated from the mixture (15,000 rpm for 20 min) and washed by DI water several times. Then, the hbPGs-*g*-UCNP-GFP sample was frozen-dry for the next biological experiment. Synthesis of hbPGs-*g*-UCNP-NH_2_ was employed except that GPF was replaced by diaethylamine.

### 2.5. NMR Titration

The NMR measurement (600MHz Nuclear Magnetic Resonance, instrumental analysis center, Shanghai Jiaotong University, Shanghai, China) was performed as follows: for every NMR tube, 20 mg of hbPGs was dissolved in 500 μL D_2_O, which contained the NaIO_4_ (5, 10, 20, 40 mg, respectively). As a control group, we also prepared the NaIO_4_ (20 mg) solutions in 500 μL.

### 2.6. Dynamic Light Scattering (DLS) and Zeta Potential

DLS and zeta potential were performed by using Malvern NanoZS (ZETASIZER Nano ZS, Malvern, Cambridge, United Kingdom) with polystyrene latex parameters at 25 °C and the measurement angle is 173°. Other parameters were set as default.

### 2.7. Cell Culture and Cell Cytotoxicity

The human lung cancer A549 and glioma U87MG cell lines were purchased from the cell bank of the Chinese Academy of Science (Shanghai, China). These cell lines were cultured in Dulbecco’s modified eagle medium (DMEM) with 10% Fetal Bovine Serum (FBS) and 100 U/L antibiotics (penicillin and streptomycin) under a humidified atmosphere including 5% CO_2_ at 37 °C. To characterize the cell cytotoxicity of polymers, cells were seeded into a 96-well with a density of 1 × 10^4^ cells/well for 200 μL, and then cells were cultured for 24 h. Polymers or nanoparticles were dissolved in the DMEM medium at different concentrations (500, 250, 125, 62.5, 31.3, 15.6, 7.8, 3.9 and 2.0 μg/mL), and 100 μL of polymer or nanoparticle solution was added to the 96-well plates after the old medium was removed. Cells were incubated with polymer or nanoparticle solutions for another 24 h and then tested by using the MTT method. The optical density (OD) value was measured at 570 nm by using a Microplate reader (Molecular Devices, San Jose, CA, USA).

### 2.8. Drug-Loading Efficiency (DLE) and Drug Release Experiment

hbPGs-Arg was suspended in DI water at a concentration of 1.0 mg/mL. Then, 0.1-mg/mL DOX aqueous solution (190 and 380 μL, respectively) was added to the suspended solution. This mixture was incubated under shaking conditions (200 rpm) at 37 °C for 24 h. After incubation finished, the suspended solution was centrifugated to obtain the top clear solution, which was used for measuring the mount of free DOX. The DOX concentration was obtained from the standard curve (V_OD_ = 0.00386 × n_c_ + 0.0389), which was built by using UV–vis method (Microplate reader, Molecular Devices, San Jose, CA, USA). Calculation of the DLE value obeys the literature reported [25].

### 2.9. Confocal Laser Scanning Microscopy (CLSM) and Inverted Fluorescence Microscopy

The CLSM sample was prepared as follows: A549 cells were seeded into a confocal dish with a glass dish on the bottom. After a 12-h incubation at 37 °C, the culture medium was removed and fresh medium containing hbPGs-*g*-UCNP-GFP (50 μg/mL) replaced it, and then it was further cultured for another 2 h. The tested samples were washed by Phosphate-buffered saline (PBS) three times to remove free hbPGs-*g*-UCNP-GFP and stained by Hoechst 33,342 for 10 min. After the nucleus staining process, cells were fixed by using 4% paraformaldehyde PBS solution for 10 min and washed with PBS to completely remove extra paraformaldehyde. These samples were covered by 3 mL of PBS solution and were immediately tested by CLSM.

The inverted fluorescence microscopy samples were prepared as follows: A549 cells was seeded into 24-well and cultured at 37 °C for 12 h. Then, old medium was removed and fresh medium with hbPGs-*g*-UCNP-GFP (10, 50, 100 and 500 μg/mL) replaced it. These samples were incubated at 37 °C for 2 h and then washed by PBS three times. All the wells were filled by 1 mL of PBS solution to be detected by using inverted fluorescence microscopy.

## 3. Results and Discussion

### 3.1. Synthesis of Hyperbranched Polyglycerols (hbPGs)

HbPGs were synthesized by using anion ring-opening multiple-branched polymerization in anhydrous dioxane [26]. Then, the crude hbPGs were purified by using a dialysis bag (molecular weight cutoff > 3.5 kDa) to remove free low-weight molecules and then frozen-dried to obtain gel-like products.

Firstly, the molecular structure was characterized by using ^1^H NMR to check the spread of hydrogen atoms (as show in Figure 1). The main backbone of hbPGs is located between 3.0 and 4.0 ppm, which is attributed to the etheric skeleton. In the expending region (SI Appendix A), the weak aromatic signal was direct observed between 7.0 and 8.0 ppm. The aromatic signal originated from the initiating core (hydroquinone moiety).

Then, the abundance of reactive *T* units was further explored by NMR techniques. To our knowledge, NaIO_4_ is well-known as vicinal diols cleavage reagent, by which aldehyde-terminated hbPGs can be easily obtained. In order to examine the ratio of reactive *T* units in hbPGs, an NaIO_4_ titration was performed as shown in Appendix A. When the mount of NaIO_4_ increased to 10 mg, the strength of aldehyde peak did not continue to be increased, which indicated that all the *T* units reacted with NaIO_4_ salt. Through the calculation process, the *T* units of hbPGs are 46.8 μmol per 20-mg weight. As a result, the weight ratio of NaIO_4_ with hbPGs is approximately 1:2 to completely obtain an aldehyde-terminated hbPGs skeleton.

As shown in Figure 1, more new peaks of well-oxidized hbPGs will be generated and several peaks will be cleared. In comparison to hbPGs and oxidized hbPGs, the new peak 5 was attributed to the hydrogen signal of an aldehyde group. We can also observe several weak peaks between 3.0 and 5.5 ppm (peaks a, b, c and d in Appendix A). Through the calculus of relative peaks, we found that the area ratio between peak a and the etheric part is approximately 18:1. Moreover, the peak groups a, b and c will be gone after reacting with ethylenediamine, which demonstrated that these peaks were related to a reductive amination process.

In the meantime, we can also observe the disappearance of peak 3, and the decrease in peak 3 is associated with the concentration of NaIO_4_. Consequently, peak 3 was attributed to the exterior *T* units of hbPGs, which is consistent with our previous report [27]. Moreover, a small chemical shift differential between peaks 1 and 4 was observed. Through the discussion below (Appendix A), we found that peak 4 was the mixed condition of repeating units (*T*, *D* or *L* units) which directly emerges in solvent. Peak a (5.08 ppm, as shown in Appendix A) was attributed to the methenyl group neighbor the aldehyde group (mechanism as shown in Appendix A). We also calculated that the oxidizing abundance is approximately 12.05% (calculated from unit ratio: 18.23/5:1/2), which indicates that only exterior *T* units can be oxidized.

Moreover, the hydrodynamic radius of hbPGs is another marker to characterize the reaction condition. Firstly, we measured the hydrodynamic radius of hbPGs in different solvents (Appendix A), which demonstrated that the distribution of hbPGs is uniform and under unimolecular condition. Then, one new molecular cluster (around 300 nm) was observed after NaIO_4_ was added to the hbPGs solution, which implied that aldehyde group can improve to the formation of cross-linking. This result is consistent with the NMR result, and it may be the main reason for aggregation after hbPG-capped nanoparticle surface engineering.

### 3.2. Exploring the Modification Mechanism of hbPGs to React with Amino Acids

To explore the modifying mechanism, we utilized amino, instead of proteins, to identify a potential modification method. Arginine is one kind of naturally necessary amino acid and a guanidyl group with a positive charge can strongly interact with phosphate ions, which plays an essential role in the regulation of nanointerfaces’ properties—for example, as gene delivery carriers [5]. As a result, quickly grafting amino acids, especially for arginine, can significantly improve the targetability and biomolecular absorption of hbPGs. Here, we utilized the aldehyde-mediated intermediator hbPG-CHO to obtain Arg-tagged hbPGs following the aforementioned method (as shown in Appendix A).

After the synthesis and purification process, we found that the hbPGs-Arg molecules cannot be dissolved by any solvent again, which is a normal phenomenon for surface engineering of high-molecular weight hbPGs, especially for introducing mount-of-surface functional groups. Consequently, exploring the molecular properties of surface-tagged hbPGs will help us to understand the especial properties of high-molecular weight hbPGs.

Firstly, IR was employed to characterize hbPG-Arg, as shown in Figure 2. In comparison to hbPGs and hbPGs-Arg compounds, the typical absorption peaks, CH stretching vibration (2872 cm^−1^ and 2915 cm^−1^) and etheric bond (1100 cm^−1^), can be significantly observed in hbPGs-Arg, which confirmed that the hbPGs skeleton has been reserved in Arg-hbPGs. Then, compared with hbPGs—arginine and hbPGs-Arg at 1637 cm^−1^—the strengthened peak (1637 cm^−1^) of hbPGs-Arg is an overlapping signal of hbPGs (1637 cm^−1^) and arginine triple peaks (1558, 1626 and 1683 cm^−1^). Moreover, the relative strength ratio between 1626 and 1100 cm^−1^ significantly increased, which confirmed that arginine molecules are successfully bonded to hbPG’s molecular skeleton.

Second, suspension of hbPGs in aqueous solution was characterized by using dynamic light scattering. When hbPGs-Arg was suspended in aqueous solution, this powder could be very easily formed in the uniformed suspended solution (size ~0.5 μm as shown in Appendix A), although this suspension will generate precipitation after several minutes. However, then, the zeta potential of hbPGs-Arg was measured in aqueous solution and showed that it is approximate 6.00 ± 0.05 mV (as shown in Appendix A), which indicated that the hbPGs-Arg surface carries a positive charge and has potential to be taken up by cells through an endocytosis process.

In order to examine the potential application of hbPGs-Arg in biomedicine, we characterized the corresponding encapsulation and release properties. Firstly, cytotoxicity by using the MTT method and hemolysis testing of hbPGs were measured as shown in Figure 2b and Appendix A, which demonstrated the low cytotoxicity of arginine-modified hbPGs even at 500 μg/mL. Then, the drug-loading efficiency (DLE) was explored by using incubation with DOX (5% and 10% weight ratio) after 24 h, which showed that DLE value was 95.83 ± 0.18% and significantly higher than other dendritic polymers. The zeta potential of hbPGs-Arg@DOX was measured as shown in SI Appendix A. The significant increase in the zeta potential (33.93 ± 1.14 mV) also confirmed that DOX molecules were successfully encapsulated by hbPGs-Arg. At last, we measured the release mount of hbPGs-Arg@DOX in a different buffer solution (pH = 5.0, 6.4, 7.4) for 24 h, and the 19.64% of DOX in acidic buffer (pH = 5.0) was released from the encapsulated complex. Moreover, there was only 1.05% released in weak acidic buffer (pH = 6.4) and a trace amount of DOX was released in neutral buffer (pH = 7.4). Moreover, we also observed that the hydrodynamic radius of DOX-loaded hbPGs-Arg decreased to ~300 nm (Appendix A). These results confirmed that arginine-modified hbPGs can be as pH-sensitive drug delivery materials to deliver guest drugs.

### 3.3. HbPGs Grafting UCNP Nanoparticles (hbPGs-g-UCNP-GFP)

Lanthanide-doped up-conversion nanoparticles (UCNPs) are a highly biocompatible and inorganic material, which has a narrow emission spectrum width, higher tissue penetration ability and low toxicity. Owing to their excellent use in biomedicine, developing the excellent water-dispersible UCNPs is of importance for further applications. The literature has previously reported [13] the use of hbPGs as a capping layer to obtain water-dispersible UCNPs (prepared in Appendix A). However, how to further modify hbPG-grafted nanoparticles is not widely reported. Here, we further functionalized hbPGs-*g*-UCNP by using our method.

Firstly, we tried to graft a GFP protein (prepared in SI) onto the hbPGs-*g*-UCNP surface. However, the hbPGs-g-UCNP-GPF with a special dealing process generated the significant aggregation as shown in Appendix A, which is hard to re-suspended in PBS solution again. As previous literature report, the dry aldehyde-terminated hbPGs will generate a gel-like solid and are hard to be dissolved in any solvent, which is owed to the acetal formation between aldehyde and hydroxyl. As a result, it is necessary to break the acetal bond by using acid—for example, HOAc or HCl. When hbPGs-g-UCNP reacts with NaIO_4_, the aggregation will generate, especially for 19 h (as shown in Appendix A). If we added the HOAc to regulate the pH of reaction system, the particle size of hbPGs-g-UCNP would not change. However, if excess HCl was added to the reaction system, particles’ size would decrease to ~200 nm, although the extended reaction time or separation–redissolve process will not change the suspension condition anymore. As a result, we employed this reaction condition to obtain protein-labeled hbPGs-*g*-UCNP-GPF particles.

HbPG-grafted UCNP (UCNP-*g*-hbPGs) was modified to obtain amino-terminated nanoparticles. Through overlaying the IR curves of hbPGs, UCNP and UCNP-NH_2_, we can clearly observe an etheric bond vibration (1107 cm^−1^) and C-C vibration (1645 cm^−1^). Presence of these peaks indicated that hbPGs are successfully grafted onto the UCNP surface and the NaIO_4_ oxidation process will not be of great impact on the stability of the hbPG layer. Moreover, the aldehyde-terminated UCNP was employed to graft biomolecules—for example, green fluorescent protein (GFP). In comparison to UCNP-GPF, hbPGs and UCNP, we can observe one new peak at 1744 cm^−1^ which was attributed to the GFP protein-related signals.

Then, we tested the cytotoxicity of hbPGs-g-UCNP-GPF in A549 cells, which is showed in Figure 3b. Similar with hbPGs-Arg, the hbPGs-g-UCNP-GPF do not have any significant impact on A549 cells even at 500 μg/mL, indicating that hbPGs can significantly improve the biocompatibility of nanoparticles.

At last, we tested the cellular uptake of hbPGs-g-UCNP-GPF by A549 (Appendix A) through confocal laser scanning microscopy, as shown in Figure 3c–e. As shown in Figure 3c, the cell nucleus of A549 was stained by Hoechst dyes, and the ball-like nucleus indicated that A549 cells were under healthy conditions, especially for inserted cell image. Through Figure 3c,d, as shown in Figure 3e, we can find that the nanoparticles were taken up by A549 cells and were uniformly distributed in the cells. Therefore, protein-modified UCNPs can be transported into the cytoplasm, which shows increased potential for them to be further applied in the biomaterials field.

## 4. Conclusions

In summary, we investigated the chemical structure of aldehyde-terminated hbPGs in aqueous solution by an NMR technique. The unique aldehyde peak indicated that the hemiacetal structure did not form after an in situ oxidation process. We utilized aldehyde-terminated hbPGs to synthesize the biomolecule-labeled hbPGs—i.e., Arg-hbPGs and hbPGs-g-UCNP-GPF. Through evaluating the cytotoxicity on A549 cells, we found that these nanocarriers have excellent biocompatibility, which can be further applied in drug delivery and tumor issue imaging. These results emphasize that aldehyde-mediated surface engineering processes on the hbPGs or hbPG-capped nanomaterials will be more efficient to be applied in biomaterials fields.

## 5. Future Direction

Owing to the wide spectrum of excitation wavelengths by UCNPs, the UCNPs displayed their potential to be applied in real-time diagnoses and disease therapy. In our study, we developed a robust and feasible UCNP modification method and significantly improved the solubility after hbPG grafting. The abundance of diol groups in hbPGs provides the modification site for carrying diagnostic and therapeutic moieties—for example, magnetic resonance imaging (MRI) contrast reagents, photodynamic therapy agents and even antibodies.

## Figures and Tables

**Figure 1 polymers-12-02592-f001:**
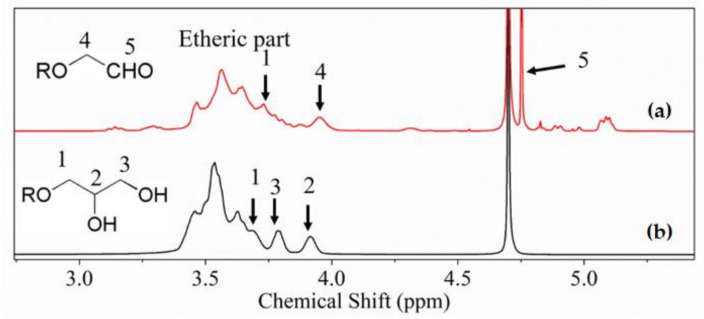
^1^H NMR spectra of hyperbranched polyglycerols (hbPGs) before (**b**) and after (**a**) reacting with NaIO_4_ in D_2_O.

**Figure 2 polymers-12-02592-f002:**
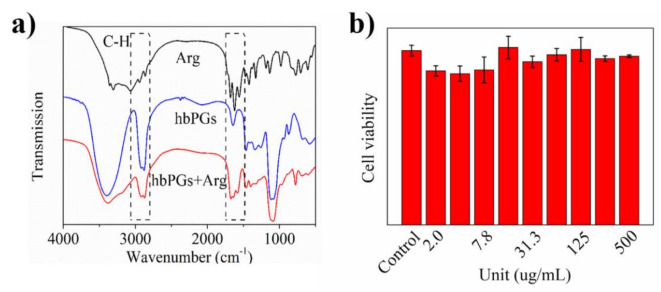
(**a**). Overlapping IR spectra of Arg, hbPGs and hbPGs-Arg. (**b**). A549 cell viability of hbPGs-Arg at different concentration (from 500 to 2.0 μg/mL).

**Figure 3 polymers-12-02592-f003:**
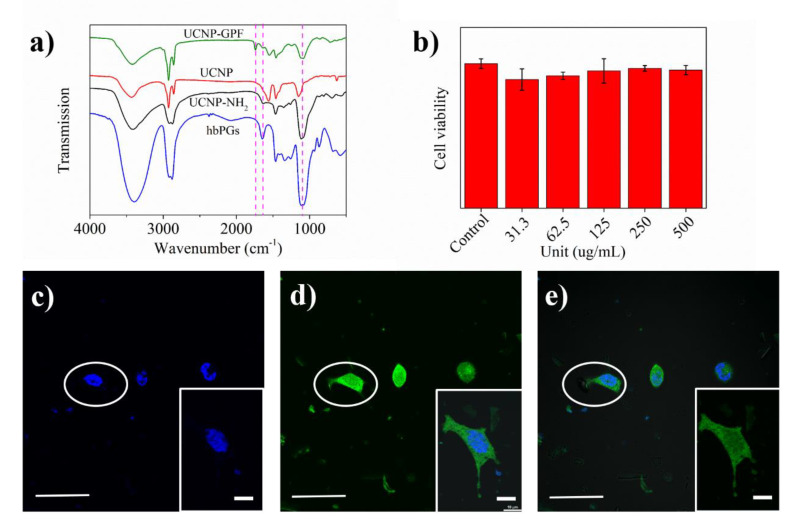
(**a**). Overlay IR spectra of hbPGs (Blue), up-conversion nanoparticles (UCNPs) (Red), UCNP-NH_2_ (Black) and UCNP-green fluorescence protein (GFP) (Green). (**b**). A549 cell viability of hbPGs-g-UCNP-GFP from 500 to 31.3 ug/mL. (**c**–**e**). Confocal images of A549 incubated with hbPGs-*g*-UCNP-GFP. Bar scale is 10 and 5 μm (inserted images), respectively. By overlapping the different staining images (in the white circles and the rectangle), the GFP-labeled UCNP is well distributed in the cytoplasm.

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
