# Peer review of "Hyperbranched Polyglycerols as Robust Up-Conversion Nanoparticle Coating Layer for Feasible Cell Imaging"

_polymers, 2020, doi:10.3390/polym12112592_

Round 1

Reviewer 1 Report

The manuscript entitled “Hyperbranched polyglycerols as robust up-conversion nanoparticle coating layer for feasible cell imaging” deals with the synthesis of hyperbranched polyglycerol (HbPG) by using hydroquinone and UCNP as initiators. The obtained products were modified to obtain aldehyde multifunctional HbPGs which were further functionalized by arginine or a GFP protein. The synthesis and characterization are well presented. The obtained results are interesting and prove the potential applications in drug delivery and bioimaging fields. However, there are still some issues to be addressed before consideration of the manuscript to be published in Polymers.

1) The preparation methods of GFP protein and UCNP are described in Supporting Information but this is not mentioned in the Manuscript. All used materials (e.g. acids and diaethylamine) should be mentioned in Chemicals and materials section and please add a sentence to this section which call the Readers attention that, GFP protein and UCNP are self-prepared, and the methods are described in the Supporting Information.

2) In my opinion, the first sentence of the Abstract is completely redundant. It only results in unnecessary repetition.

3) In the Introduction the application of the nano-materials are mentioned, but there is not a good paragraph which deals with the UCPNs. However, one example is presented (PEG-grafted UCNP) but the application of UCNP-s is highly important, therefore, one short paragraph related to the UCNPs should be added to the Introduction.

4) The "2.4 Synthesis of hbPGs-g-UCNP" heading is incomplete because here the two-step reductive amination is also detailed. The section heading should be revised, or the synthetic steps should be edited under separate headings.

5) The Authors synthesized a high molecular weight HbPG. What is the molecular weight of the HbPG? This can be approximated by the integral ratio of the hydroxyl protons and aromatic protons in the 1H spectrum of hbPGs measured in d6-DMSO (Fig S1).

6) The Authors stated in the Conclusion that hemiacetal is not formed during the oxidation. But the size of the nanoparticles measured by DLS show a "new molecular claster" formation, which was explained by cross-linking formation. Morovere, in case of the synthesis of hBPG-g-UCNP-GFP, acidic conditions were tested to reduce the size of the nanoparticles. Please clarify the contradiction.

7) One of the most interesting and important part of the Manuscript is the drug encapsulation and release properties. As written the size is decreased after release but Fig S8 shows the hydrodynamic radius profile of hbPGs-Arg before (a) and after (b) encapsulating DOX molecules. Why does DOX encapsulation decrease the size of the nanoparticle as shown in Fig S8? Please clarify the contradiction.

8) Why does encapsulated DOX increase the zeta potential of the nanoparticle? Does it not primarily depend on the surface charge of the particle?

9) In Figure 2 the assignation is incomplete. The sign '2' is missing on the spectrum and '4' is missing on the structure.

10) Figure S5 is not mentioned in the Manuscript.

11) The English of the manuscript should be checked. Some examples:

- line 73: Therefore to then or thereafter

- line 105: solution to solutionS

- line 139: “The inverted fluorescence microscopy samples as fallows:” missing verb

etc.

Author Response

Dear reviewer,

 We have up-loaded the response letter. Please see the attachment.

Reviewer 2 Report

Hao et al reported the synthesis and use of hyperbranched polyglycerols for coating of upconversion nanoparticles for cell imaging. This study is generally quite interesting and may be of interest to the readers of Polymers. However, the current manuscript suffers from several issues that need to be addressed before it could be reconsidered for publication.

  1. The authors should justify the selection of hyperbranched polyglycerols as the coating layer as compared to other polymeric materials (for example, polyethylene glycol (PEG)). Other polymeric coatings have also been shown to endow the coated nanoparticles with excellent water dispersibility and biocompatibility. More concrete information on this should be included in the manuscript.
  2. Figure 1 is missing from the manuscript.
  3. How is the stability of UCNP, hbPGs-g-UCNP, and hbPGs-g-UCNP-GFP in different solutions over time? The authors should characterize and provide this information in the manuscript.
  4. For Figure 4, it is unclear what panels c, d, and e refer to. Similarly, it is unclear which scale bars are 20 μm and 5 μm.
  5. How is the biocompatibility of hbPGs-g-UCNP with respect to healthy cells? The authors are encouraged to test the cytotoxicity of hbPGs-g-UCNP on healthy cells instead of on cancer cells.
  6. The authors should also characterize the toxicity of bare UCNP and compare this to the toxicity of hbPGs-g-UCNP and hbPGs-g-UCNP-GFP to show that the polyglycerol coating indeed improves the biocompatibility of the nanoparticles.
  7. The authors are encouraged to have the manuscript proofread and revised by someone with an acceptable level of English language proficiency. The current manuscript suffers from many grammatical and spelling errors.

Round 2

Reviewer 2 Report

The authors have largely addressed most of my comments. I would suggest to include Figure RS2 in the manuscript or supporting information of the manuscript.
